# Invasive pneumococcal disease in Latin America and the Caribbean: Serotype distribution, disease burden, and impact of vaccination. A systematic review and meta-analysis

Ariel Bardach[1,2]*, Silvina Ruvinsky[3,4], M. Carolina Palermo[1], Tomás Alconada[1], M. Macarena Sandoval[1], Martín E. Brizuela[5], Eugenia Ramirez Wierzbicki[1], Joaquín Cantos[1], Paula Gagetti[6], Agustín Ciapponi[1,2]

1 Instituto de Efectividad Clínica y Sanitaria (IECS-CONICET), Buenos Aires, Argentina, 2 Centro de Investigaciones Epidemiológicas y Salud Pública (CIESP-IECS), CONICET, Buenos Aires, Argentina, 3 Departamento de Investigación, Hospital Garrahan, Buenos Aires, Argentina, 4 Departamento de Evaluación de Tecnologías Sanitarias y Economía de la Salud, Instituto de Efectividad Clínica y Sanitaria, Buenos Aires, Argentina, 5 Unidad de Pediatría, Hospital General de Agudos Vélez Sarsfield, Buenos Aires, Argentina, 6 Servicio Antimicrobianos, Instituto Nacional de Enfermedades Infecciosas (INEI)-ANLIS "Dr. Carlos G. Malbrán", Buenos Aires, Argentina

* abardach@iecs.org.ar

## Abstract

### Background

Invasive pneumococcal diseases (IPD) are associated with high morbidity, mortality, and health costs worldwide, particularly in Latin America and the Caribbean (LAC). Surveillance about the distribution of serotypes causing IPD and the impact of pneumococcal vaccination is an important epidemiological tool to monitor disease activity trends, inform public health decision-making, and implement relevant prevention and control measures.

### Objectives

To estimate the serotype distribution for IPD and the related disease burden in LAC before, during, and after implementing the pneumococcal vaccine immunization program in LAC.

### Methods

Systematic literature review following Cochrane methods of studies from LAC. We evaluated the impact of the pneumococcal vaccine on hospitalization and death during or after hospitalizations due to pneumococcal disease and serotype-specific disease over time. We also analyzed the incidence of serotyped IPD in pneumococcal conjugate vaccine PCV10 and PCV13. The protocol was registered in PROSPERO (ID: CRD42023392097).

**Data Availability Statement:** All relevant data are within the manuscript and its Supporting information files.

**Funding:** Pfizer Independent Grant 76436251. The funders had no role in study design, data collection and analysis, decision to publish, or preparation of the manuscript.

**Competing interests:** The authors have declared that no competing interests exist.

## Results

155 epidemiological studies were screened and provided epidemiological data on IPD. Meta-analysis of invasive diseases in children <5 years old found that 57%-65% of causative serotypes were included in PCV10 and 66%-84% in PCV13. After PCV introduction, vaccine serotypes declined in IPD, and the emergence of non-vaccine serotypes varied by country.

## Conclusions

Pneumococcal conjugate vaccines significantly reduced IPD and shifted serotype distribution in Latin America and the Caribbean. PCV10/PCV13 covered 57–84% of serotypes in children under 5, with marked decline in PCV serotypes post-vaccination. Continuous surveillance remains crucial for monitoring evolving serotypes and informing public health action.

## Introduction

*Streptococcus pneumoniae* is a main cause of infections globally producing different clinical manifestations. Invasive pneumococcal diseases (IPD), including meningitis, bacteremic pneumonia, and other clinical manifestations with pneumococcal isolated from sterile sites, are associated with high morbidity, mortality, and high public health impact worldwide [1, 2]. *S. pneumoniae* is the leading cause of bacterial pneumonia producing higher morbidity and mortality among children under five years from low and middle income countries (LMIC) [3, 4]. Meningitis causes more than 2.5 million new cases globally and 230,000 deaths and *S. pneumoniae* causes the largest proportion of total all-ages meningitis deaths [5]. On the other hand, is necessary to highlight that immunocompromised hosts are at increased risk of IPD [6].

The Pan American Health Organization (PAHO) in 2011 recommended the introduction of PCVs into the National Immunization Programs (NIP) of countries in the Latin America and the Caribbean (LAC) region. Since May 2016, 29 LAC countries have been using PCV10 or PCV13 in their countries [7] with different scheduling strategies [8–10].

Pneumococcal serotype distribution varies by age, clinical manifestation, period, and geographical region [11]. Infectious disease surveillance is an important tool for monitoring trends and epidemiology of the disease and generating real-time information for decision-makers before introducing new initiatives for pneumococcal preventable diseases in the LAC region [12]. The Regional Immunization System or Sistema Regional de Vacunas in Spanish (SIREVA) is a network of national reference laboratories conducting surveillance of bacterial pneumonia, sepsis/bacteremia, and meningitis in Latin America since 1993 and is organized by PAHO [12]. The SIREVA II project was launched in 2005 by PAHO to improve regional disease surveillance with a total of national reference laboratories in 19 countries [8, 13]. Routine implementation of pneumococcal conjugate vaccines (PCV7/10/13) in immunization programs has significantly reduced IPD produced by serotypes included in these vaccines [14]. On the other hand, PCV has been associated with a decrease in IPD due to vaccine serotypes among the unvaccinated population due to herd immunity [15]. However, non PCV serotypes were found related to IPD in children. This systematic review and meta-analysis aims to improve knowledge about the IPD burden, serotype distribution, and impact of PCVs over the last two decades in the LAC region.

## Material and methods

We conducted this systematic literature review of international, regional, and country-published and unpublished data, together with reports of routinely recorded data such as registries and MoH, following Cochrane methods [16].

We performed a meta-analysis following the MOOSE guidelines for observational studies [17] and the PRISMA statement for reporting systematic reviews and meta-analyses [18]. The protocol followed the PRISMA-P [19] statement and was registered in PROSPERO, the International Prospective Register of Systematic Reviews (ID: CRD42023392097).

### Eligibility criteria

We included cohort studies, case-control, cross-sectional studies, epidemiological surveillance reports, hospital-based surveillance studies, case series, control arms of randomized/quasi-randomized controlled trials, controlled before and after (CBA) and uncontrolled before and after (UBA) studies, interrupted time series (ITS), and controlled ITS (CITS), assessing serotyping of IPD cases. IPD clinical presentations include sepsis/bacteremia, meningitis, or bacteremic pneumonia, and other presentations of IPD, such as empyema, peritonitis, osteoarticular infection / septic arthritis, or endocarditis. No language restriction was imposed. Studies with at least 20 culture-confirmed cases from typically sterile sites (e.g., blood, cerebrospinal fluid, pleural fluid) published or reported since January 2000 were included. Every LAC country represented the geographical scope.

Systematic reviews and meta-analyses were considered a source of primary studies. In those cases where data or data subsets were reported in more than one publication, the one with the larger sample size was selected. All LAC patients were susceptible or suffering from probable or confirmed cases of IPD, belonging or not to risk groups.

### Data sources and search strategy

We systematically search the primary literature, international and regional databases, generic and academic internet searches, and meta-search engines. We searched records from 1 Jan 2000 to 31 Dec 2022 from the following databases: MEDLINE (PubMed), EMBASE (Elsevier interface), LILACS/ Scielo, EconLIT (EBSCO Interface), Global Health (OVID), CINAHL (EBSCO Interface), and Web of Science. The SIREVA and other surveillance databases were searched. Other databases containing regional proceedings, congresses' annals, and doctoral theses were also searched. We also consult websites from the leading regional medical societies, experts, and associations related to the topic. An annotated search strategy for grey literature was included to retrieve information from relevant sources like regional MoH, PAHO, and reports from hospitals. This strategy comprises two blocks plus search limits (from 2000 onwards to cover ten years of pre-vaccination programs) and a geographical string block for LAC. The search had no language restrictions and was limited to humans. The reference lists of the articles were hand-searched for additional information. After consulting the principal investigator, we selected only the publication with the largest sample size. Highly cited authors were contacted to obtain missing or extra information.

### Selection of studies, data extraction, and assessment of the risk of bias

Potentially eligible studies were retrieved in full-text for further analysis independently by two researchers. Any disagreement was resolved by discussion between the two reviewers. Discrepancies were solved by consensus of the whole work team. We utilized the COVIDENCE® software [20, 21]. This form was piloted on ten papers to refine the process. We also retrieved

studies that would add to the general discussion or added covariates potentially useful for the analysis.

The risk of bias in observational studies and the control arm of trials was assessed using the checklists developed by the United States National Heart, Lung, and Blood Institute, which classify studies as high risk of bias (Poor), moderate risk of bias (Fair), and low risk of bias (Good). For the assessment of cohort studies and cross-sectional studies, the tool comprises 14 items, while nine items apply to the case series studies. For RCTs and quasi-RCTs, we evaluated the following domains: sequence generation, allocation concealment, blinding of participants and personnel, blinding of outcome assessors, incomplete outcome data, selective outcome reporting, and other possible threats to validity [22]. For interrupted time series (ITSs): intervention independent of other changes; pre-specification of the form of the intervention effect; the likelihood that the intervention will affect data collection; blinding to intervention allocation of outcome assessors; incomplete outcome data; selective reporting of results; and other sources of bias [23]. Extraction was performed in a prepiloted spreadsheet by independent reviewers.

We classified each criterion as "low risk," "high risk," and "uncertain risk," together with a descriptive summary of the information that influenced our judgment. When the criteria are rated "uncertain," we obtained more information from the study authors. Pairs of independent reviewers assessed the risk of bias. Discrepancies were solved by consensus of the whole team.

## Data synthesis

**Primary análisis.** To analyze our data, we conducted a proportion meta-analysis, which was performed using the metaprop {meta} package with R software version 4.2.2 [24, 25]. We applied an arc-sine transformation to stabilize the variance of proportions (Freeman-Tukey variant of the arc-sine square-root of transformed proportions method) [26]. The pooled proportion was calculated as the back-transformation of the weighted mean of the transformed proportions, using inverse arcsine variance weights for the fixed and random effects model. We applied DerSimonian-Laird weights for the random effects model where heterogeneity between studies was found. We calculated the $I^2$ statistic as a measure of the proportion of the overall variation in the proportion attributable to between-study heterogeneity. An $I^2$ >60–70% was considered substantial heterogeneity, and below 30% was a low level of heterogeneity [27].

Because the follow-up period of the studies varied considerably, we based the calculation of incidence rates on person-years dividing the number of new cases occurring during the follow-up period (the numerator) by the total person-time units (person-years) of the group at risk (the denominator). The person-time incidence rate, or incidence density rate, is an appropriate measure of incidence when follow-up times are unequal. Because we had considerable heterogeneity, we performed a random-effects model. Incidence was expressed as the number of cases per 100,000 person-years.

To analyze the evolution of serotype distribution and the number of *S. pneumoniae* isolates in LAC countries from 2006 to 2018, we performed an interrupted time series (ITS) analysis using data collected from SIREVA reports. We used a linear regression model to analyze the relationship between our dependent variables (Y) and time (T), intervention (D), and time since intervention (P) for children under five years old. The model was specified as Y ~ T + D + P and was fitted using the lm() function in R software [24] with data from the dataset. This analysis aimed to assess the intervention's effect (PCV13 or PCV10 introduction as appropriate) on the serotype distribution of *S. pneumoniae* over time. We also evaluated the serotypes included in the recently approved PCV20 with the data available from 2013 to 2018. To test for

autocorrelation in the residuals of the linear regression model, we used the Durbin-Watson test implemented in the dwtest() function from lmtest() package [28]. We also plotted the autocorrelation function of the residuals using the acf() function from stats() package [24]. Based on these results, we fit a generalized least squares (GLS) model with an autoregressive moving average (ARMA) correlation structure using the gls() function from nlme() package [29]. The GLS model was specified as Y ~ T + D + P with an ARMA (p,q) correlation structure and was fitted using maximum likelihood estimation. The values of p and q for the ARMA correlation structure were determined by inspecting the autocorrelation and partial autocorrelation plots. We also used the Akaike information criterion (AIC) and Bayesian information criterion (BIC) to select the correlation structure that best represented the data.

**Subgroup analyses and investigation of heterogeneity.** We analyzed the subgroups classifying the studies by five-year calendar period, country, and age group (0–5 years, 6–64 years, 65 or more years old), by each IPD (sepsis/bacteremia, meningitis, or bacteremic pneumonia). We included data on different *S. pneumoniae* serotypes obtained from normally sterile sites (e.g., blood, cerebrospinal fluid, pleural fluid). Both types of analyses could contribute to the investigation of heterogeneity causes.

## Results

We identified 8,600 records in seven literature databases. Of those registries, 427 were evaluated in full text to determine their eligibility. Finally, 155 studies [30–184] met the inclusion criteria. Fig 1 **PRISMA flow diagram and** S1 Checklist. The search for gray literature was performed using the following sources: generic internet search and metasearch engines (Google), and surveillance data from SIREVA were also obtained. The search strategy is available in S1 File, and the list of studies excluded during the extraction process is in S3 Table 1 in S2 File.

### Characteristics of included studies

We included 155 studies [30–184] with data from 170,054 patients. Eighty-nine studies provide data on IPD (n = 136,276), 33 on meningitis (n = 19,032), 29 on pneumonia (n = 16,172), and four on bacteremia (n = 398). The study descriptions are shown in the S3 Table 1. **Characteristics of included studies** of the S2 File. The studies were carried out in 18 countries; the most represented countries were Brazil (39), Argentina (26), Colombia (17), and Uruguay (15). There were also six multi-country studies.

Regarding the methodological design, there were 83 cross-sectional studies, including 41 surveillance studies, 61 case series, eight cohort studies, and three ecological studies. As for the characteristics of the identified participants, 88 (57%) studies included pediatric patients only (n = 99,067), 23 (15%) only adults (n = 4,768), and 41 (26%) both pediatric and adult populations (n = 64,828). Only three (2%) studies did not report the participants' age (n = 1,247). All participants came from non-probability sampling. The age range was from 0 to 94 years, with a reported mean range between 10 days to 76.6 years and a median range between 3 days to 73 years.

### Main findings

In all cases analyzed, patients under five years have shown a higher proportion of IPD, followed by patients over 65 years old, and finally by age group from 6 to 64 years old.

When we analyzed serotypes, we found that in global IPD, as well as in pneumonia, bacteremia, and meningitis, more frequent serotypes were included in the PCV13. In contrast, PCV10 serotypes caused about two-thirds of IPD.

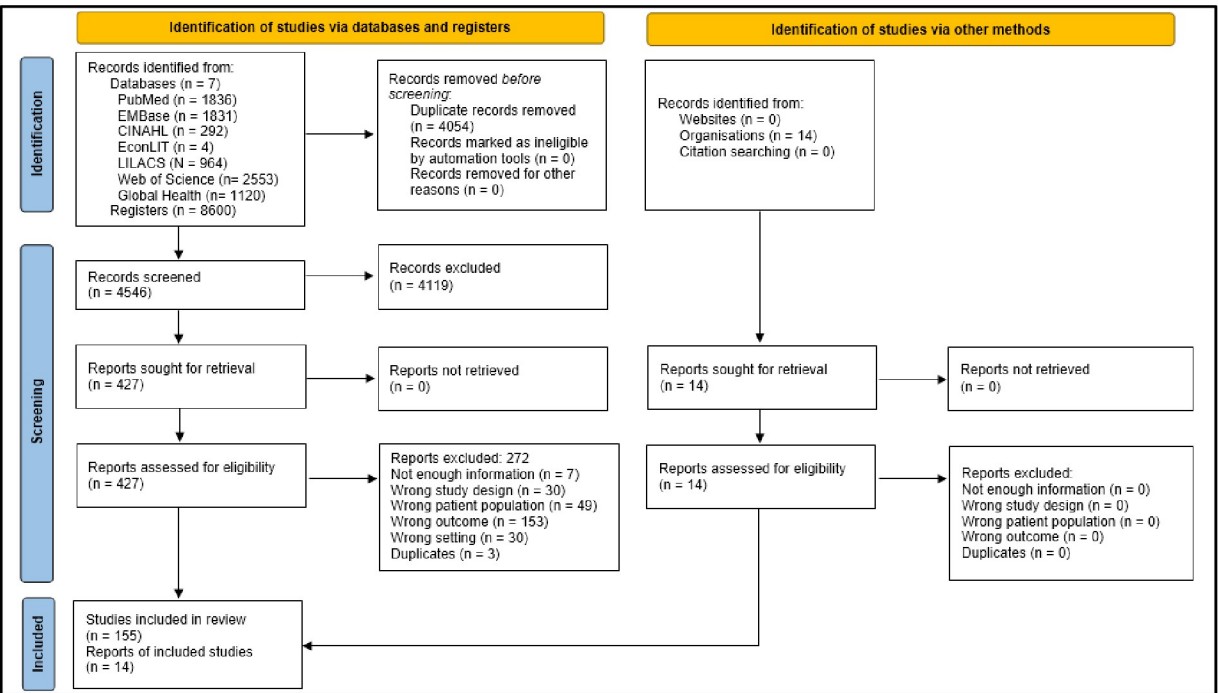

**Fig 1. PRISMA flow diagram.**

The distribution of serotypes by period shows a decreasing trend since 2010 for PCV10 serotypes, observed for every type of IPD infection. On the other hand, PCV13 serotypes have shown a stable frequency over time, with no significant differences between lustrum. We found substantial variability in the number of reports by country. Most countries that reported information about IPD and serotype-related infection were Brazil (n = 42,054 patients), Argentina (n = 34,780 patients), Colombia (n = 15,176 patients), and Chile (n = 14,138 patients), Table 1.

A meta-analysis of the incidence of IPD was performed, considering the distribution by age group. The results show that the incidence of IPD varies significantly by age. Specifically, a higher rate was observed in children under five, registering 24,29 events per 100,000 persons/ year. In contrast, the group from 6 to 64 exhibited a much lower incidence, with 2.33 events per 100,000 persons/year. The data also revealed that those older than 65 experienced an inter- mediate incidence of 8.81 events per 100,000 persons/year. This disparity in incidence rates highlights the particular vulnerability of young children and older adults to IPD compared to midlife adults. S3 Table 2 in S2 File and Fig 2 **Meta-analysis of incidence of all Invasive Pneu- mococcal Disease by age groups**.

Likewise, concerning IPD by a specific condition, from meningitis, it was only possible to perform a meta-analysis of incidence in children under five years of age, obtaining 9.28 events per 100,000 persons/year. The analysis could not be carried out in sepsis and pneumonia due to the absence of studies reporting incidence S3 Table 3 in S2 File.

Finally, the case fatality rates (CFR) of meningitis and bacteremia were the highest, with 24% and 18%, respectively. Patients >65 years had the highest lethality for all diagnoses except meningitis, where children <5 years had a lethality of 28%, followed by the 6–64 years group with 14%. When we analyzed the CFR by country, selecting those that contributed with at least

**Table 1. Proportion meta-analyses of the distribution of vaccine serotypes by condition in LAC, by five-year period, age groups, and countries.**

| Distribution of vaccine serotypes | IPD | | | | | | Pneumonia | | | | | | Meningitis | | | | | | Bacteremia | | | | | |
|---|---|---|---|---|---|---|---|---|---|---|---|---|---|---|---|---|---|---|---|---|---|---|---|---|
| | N of studies | PCV10 Proportion (CI 95%) | I2 | N of studies | PCV13 Proportion (CI 95%) | I2 | N of studies | PCV10 Proportion (CI 95%) | I2 | N of studies | PCV13 Proportion (CI 95%) | I2 | N of studies | PCV10 Proportion (CI 95%) | I2 | N of studies | PCV13 Proportion (CI 95%) | I2 | N of studies | PCV10 Proportion (CI 95%) | I2 | N of studies | PCV13 Proportion (CI 95%) | I2 |
| overall | 65 | 57.28% (49.85%-64.40%) | 98.80% | 69 | 71.68% (64.80%-77.68%) | 99.50% | 31 | 64.71% (56.52%-72.12%) | 96% | 31 | 84.16% (78.71%-88.42%) | 93.80% | 29 | 53% (44%-62%) | 92% | 29 | 66% (56%-75%) | 91% | 7 | 62.34% (36.50%-82.65%) | 97.50% | 7 | 75.98% (46.73%-91.94%) | 98.20% |
| **by 5-years period** | | | | | | | | | | | | | | | | | | | | | | | | |
| 1990-1994 | - | - | - | 2 | 3.23% (1.69%-6.08%) | 0% | - | - | - | - | - | - | - | - | - | - | - | - | - | - | - | - | - | - |
| 1995-1999 | 12 | 68.65% (49.06%-83.27%) | 98.80% | 17 | 46.57% (20.18%-75.04%) | 98.70% | 7 | 74.82% (66.18%-81.87%) | 90.50% | 7 | 80.86% (72.27%-87.26%) | 91.40% | 2 | 62% (40%-80%) | 92% | 2 | 71% (65%-77%) | 29% | - | - | - | - | - | - |
| 2000-2004 | 15 | 67.87% (53.85%-79.28%) | 98% | 21 | 55.34% (29.94%-78.23%) | 98.30% | 6 | 85.71% (83.64%-87.55%) | 0% | 6 | 94.88% (91.94%-96.78%) | 62% | 2 | 51% (36%-66%) | 0% | 2 | 66% (47%-81%) | 86% | - | - | - | - | - | - |
| 2005-2009 | 12 | 60.16% (43.62%-74.67%) | 98.80% | 19 | 47.41% (23.46%-72.61%) | 99.40% | 6 | 64.70% (48.18%-78.33%) | 89% | 6 | 81.68% (64.19%-91.73%) | 90.30% | 2 | 66% (45%-82%) | 81% | 2 | 74% (58%-%85) | 67% | 2 | 84.08% (36.58%-97.97%) | 92.90% | 2 | 94.47% (34.98%-99.82%) | 84.10% |
| 2010-2014 | 26 | 45.50% (36.35%-54.96%) | 97.90% | 33 | 51.58% (36.79%-66.09%) | 99% | 11 | 55.89% (48.09%-63.41%) | 66.80% | 11 | 79.46% (72.69%-84.91%) | 62.30% | 3 | 23% (1%-90%) | 88% | 3 | 62% (16%-93%) | 95% | 3 | 57.63% (50.60%-64.36%) | 0% | 3 | 73.40% (66.76%-79.12%) | 0% |
| 2015-2019 | 8 | 17.03% (10.40%-26.62%) | 90.40% | 10 | 52.59% (37.39%-67.32%) | 98.8% | 1 | 14.91% (10.20%-21.28%) | NA | 1 | 75.16% (67.91%-81.22%) | NA | - | - | - | - | - | - | - | - | - | - | - | - |
| **by age** | | | | | | | | | | | | | | | | | | | | | | | | |
| 0-5 years | 45 | 72.00% (67.28%-76.23%) | 93.80% | 70 | 54.77% (39.87%-68.87%) | 99.80% | 24 | 76.67% (70.59%-81.82%) | 90.30% | 24 | 87.00% (81.75%-90.90%) | 92.60% | 6 | 74% (70%-77%) | 0% | 6 | 84% (74%-90%) | 43% | 2 | 33.01% (1.88%-92.68%) | 99.30% | 2 | 40.51% (1.38%-97.07%) | 99.40% |
| 6-64 years | 14 | 52.28% (43.49%-60.93%) | 93.60% | 17 | 65.83% (57.73%-73.11%) | 90.70% | 5 | 55.24% (46.13%-64.02%) | 0% | 5 | 68.84% (59.86%-76.60%) | 0% | 1 | 30% (21%-42%) | NA | 1 | 38% (27%-50%) | NA | 1 | 7.32% (2.38%-20.37%) | NA | 1 | 12.20% (5.17%-26.14%) | NA |
| ≥ 65 years | 8 | 34.03% (24.44%-45.14%) | 86.40% | 9 | 59.38% (51.36%-66.93%) | 77.80% | - | - | - | - | - | - | - | - | - | - | - | - | - | - | - | - | - | - |
| **by country** | | | | | | | | | | | | | | | | | | | | | | | | |
| Argentina | 8 | 78.35% (71.27%-84.08%) | 91.40% | 9 | 80.13% (53.23%-93.46%) | 99.80% | 4 | 76.63% (61.96%-86.85%) | 95.30% | 4 | 83.44% (70.66%-91.33%) | 96.30% | 3 | 78% (69%-85%) | 67% | 3 | 85% (76%-91%) | 67% | 2 | 64.83% (48.81%-78.08%) | 90.3 | 2 | 78.40% (66.46%-86.92%) | 84.90% |
| Brazil | 17 | 58.30% (48.02%-67.90%) | 92.50% | 18 | 68.28% (54.85%-79.22%) | 99.70% | 5 | 76.05% (67.81%-82.71%) | 85.50% | 5 | 88.60% (79.52%-93.96%) | 82.40% | 12 | 38% (18%-63%) | 88% | 12 | 55% (32%-76%) | 92% | - | - | - | - | - | - |
| Chile | 3 | 50.55% (35.53%-65.47%) | 94.70% | 3 | 63.78% (35.80%-84.76%) | 98.00% | 2 | 51.09% (31.59%-70.26%) | 80.80% | 2 | 60.97% (34.64%-82.16%) | 88.20% | - | - | - | - | - | - | 2 | 59.78% (1.22%-99.44%) | 98.40% | 2 | 79.71% (0.51%-99.97%) | 95.60% |
| Colombia | 11 | 57.63% (28.08%-82.58%) | 99.60% | 12 | 70.68% (43.00%-88.51%) | 99.60% | 4 | 49.33% (23.32%-75.71%) | 98.50% | 4 | 72.04% (57.07%-83.32%) | 96.10% | 3 | 36% (16%-62%) | 94% | 3 | 59% (45%-72%) | 84% | 1 | 64.73% (60.10%-69.10%) | NA | 1 | 79.12% (75.02%-82.70%) | NA |
| Costa Rica | 1 | 52.38% (31.84%-72.15%) | NA | 1 | 80.95% (58.85%-92.66%) | NA | 1 | 51.85% (33.61%-69.61%) | NA | 1 | 98.21% (77.04%-99.89%) | NA | - | - | - | - | - | - | - | - | - | - | - | - |
| Cuba | 4 | 47.93% (36.29%-59.80%) | 64.40% | 4 | 83.56% (75.74%-89.21%) | 37.80% | 3 | 43.21% (35.39%-51.38%) | 0% | 3 | 84.70% (64.89%-94.31%) | 78.30% | 4 | 61% (51%-70%) | 10% | 4 | 81% (64%-91%) | 78.30% | 1 | 66.67% (26.81%-91.61%) | NA | 1 | 83.33% (36.87%-97.72%) | NA |
| Dominican-Republic | - | - | - | - | - | - | 1 | 59.67% (52.37%-66.57%) | NA | 1 | 83.43% (77.28%-88.16%) | NA | - | - | NA | - | - | NA | - | - | - | - | - | NA |
| French-Guiana | - | - | - | - | - | - | - | - | - | - | - | - | 1 | 18% (6%-37%) | NA | 1 | 18% (6%-37%) | NA | - | - | - | - | - | - |
| Guatemala | 2 | 19.62% (1.22%-82.80%) | 96.80% | 2 | 48.11% (25.21%-71.84%) | NA | - | - | - | - | - | - | - | - | - | - | - | - | - | - | - | - | - | - |
| Jamaica | 1 | 26.32% (16.53%-39.17%) | NA | 1 | 29.82% (19.42%-42.84%) | NA | - | - | - | - | - | - | - | - | - | - | - | - | - | - | - | - | - | - |
| Mexico | 6 | 34.18% (16.50%-57.72%) | 88.70% | 6 | 70.66% (66.41%-74.57%) | 34.10% | 2 | 43.09% (24.45%-63.92%) | 76.60% | 2 | 60.61% (54.02%-66.84%) | 0% | - | - | - | - | - | - | - | - | - | - | - | - |

*(Continued)*

**Table 1.** (Continued)

| Distribution of vaccine serotypes | IPD | | | | | | Pneumonia | | | | | | Meningitis | | | | | | Bacteremia | | | | | |
|---|---|---|---|---|---|---|---|---|---|---|---|---|---|---|---|---|---|---|---|---|---|---|---|---|
| | PCV10 | | | PCV13 | | | PCV10 | | | PCV13 | | | PCV10 | | | PCV13 | | | PCV10 | | | PCV13 | | |
| | N of studies | Proportion (CI 95%) | I2 | N of studies | Proportion (CI 95%) | I2 | N of studies | Proportion (CI 95%) | I2 | N of studies | Proportion (CI 95%) | I2 | N of studies | Proportion (CI 95%) | I2 | N of studies | Proportion (CI 95%) | I2 | N of studies | Proportion (CI 95%) | I2 | N of studies | Proportion (CI 95%) | I2 |
| **Panama** | 1 | 92.86% (62.97%-99.00%) | NA | 1 | 92.86% (62.97%-99.00%) | NA | - | - | - | - | - | - | - | - | - | - | - | - | - | - | - | - | - | - |
| **Paraguay** | 2 | 58.41% (24.26%-86.02%) | 96.30% | 2 | 71.45% (53.61%-84.43%) | 85.00% | - | - | - | - | - | - | 1 | 70%(51%-85%) | NA | 1 | 70%(51%-85%) | NA | - | - | - | - | - | - |
| **Peru** | 5 | 64.63% (54.76%-73.39%) | 67.20% | 6 | 75.58% (64.13%-84.28%) | 75.50% | 1 | 77.08% (63.18%-86.83%) | NA | 1 | 89.58% (77.31%-95.60%) | NA | 1 | 77%(61%-89%) | NA | 1 | 82%(66%-92%) | NA | - | - | - | - | - | - |
| **Trinidad-Tobago** | - | - | - | - | - | - | 1 | 50.00% (31.67%-68.33%) | NA | 1 | 61.54% (42.07%-77.90%) | NA | 1 | 57%(29%-82%) | NA | 1 | 86%(57%-98%) | NA | 1 | 63.41% (47.87%-76.59%) | NA | 1 | 78.05% (62.93%-88.16%) | NA |
| **Uruguay** | 4 | 62.36% (41.13%-79.71%) | 97.90% | 4 | 73.12% (46.27%-89.58%) | 98.20% | 10 | 72.26% (58.58%-82.76%) | 94.20% | 10 | 87.89% (78.03%-93.69%) | 94.10% | 3 | 55%(30%-78%) | 93% | 3 | 62%(38%-82%) | 93% | - | - | - | - | - | - |

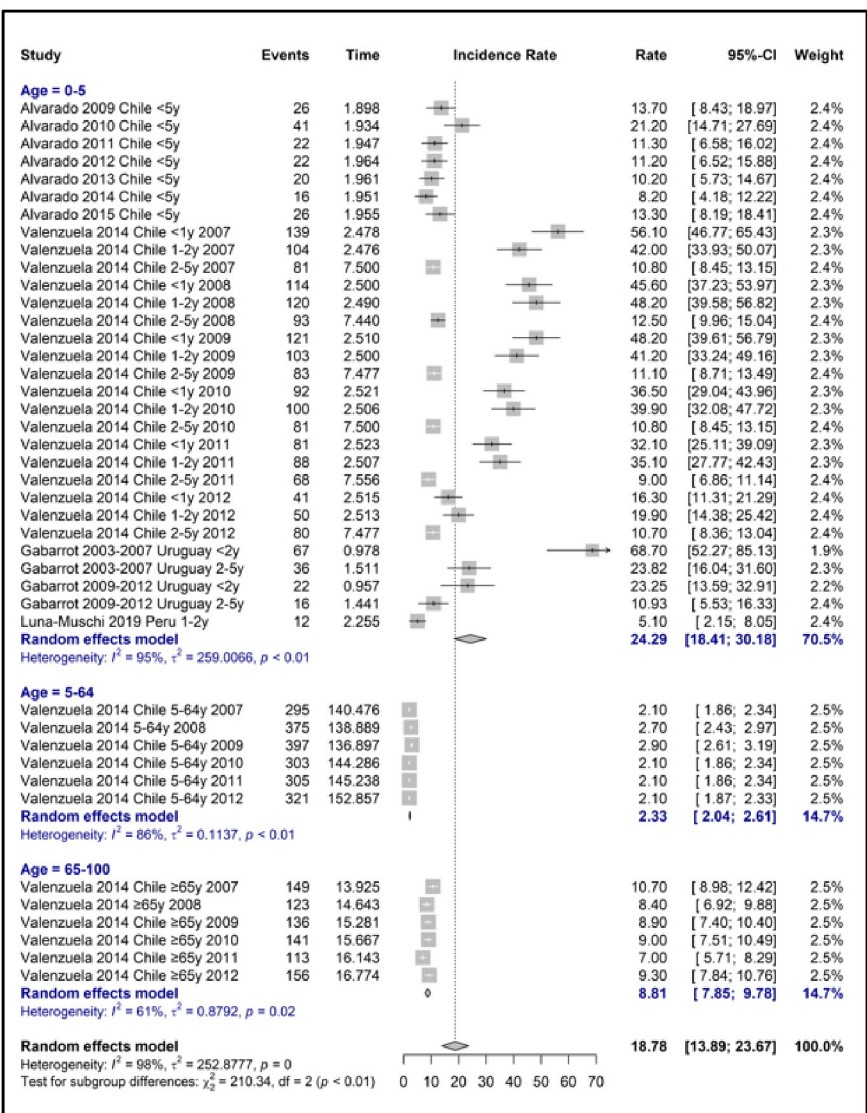

**Fig 2. Meta-analysis of incidence of all invasive pneumococcal disease by age groups.**

five studies, we found that the highest CFRs were observed in Mexico and Colombia for every type of infection, with variable magnitudes, ranging between 25% in the cases of IPD from Mexico and 38% in the cases of meningitis in the same country, Table 2.

## Trends of pneumococcal serotype distribution

SIREVA data reported 57,836 strains of *S. pneumoniae* isolated from sterile sites of patients with IPD in 19 countries from the LAC region from 2006 to 2018. According to these data, the introduction of PCVs in the national immunization programs has reduced the proportion of serotypes causing infection in countries such as Argentina, Brazil, Chile, and Colombia. These reductions show a distinct trend in some countries; Fig 3 *S. pneumoniae* **isolates before and after PCV13 and PCV10 introduction in Argentina, Brazil, Chile, and Colombia between 2006 and 2018**. It has been more remarkable in Argentina and Brazil, with no significant

**Table 2. Case fatality rate according to type of IPD and sub- analysis by 5-year period, age and country.**

| Case fatality rate (CFR) | IPD | | | Pneumonia | | | Meningitis | | | Bacteremia | | |
|---|---|---|---|---|---|---|---|---|---|---|---|---|
| | Studies number | Proportion (CI 95%) | I2 | Studies number | Proportion (CI 95%) | I2 | Studies number | Proportion (CI 95%) | I2 | Studies number | Proportion (CI 95%) | I2 |
| **Overall** | 41 | 13.83% (10.87%-17.42%) | 95.20% | 32 | 8.89% (6.17%-12.63%) | 96.50% | 39 | 24% (21%-28%) | 98% | 16 | 17.73% (12.25%-24.98%) | 90.70% |
| **by 5-years period** | | | | | | | | | | | | |
| **1990–1994** | 1 | 24.59% (17.76%-32.99%) | NA | - | - | - | - | - | - | - | - | - |
| **1995–1999** | 3 | 9.39% (5.45%-15.71%) | 71.60% | 10 | 7.12% (4.05%-12.22%) | 85.20% | | 25% (16%-38%) | 83% | 2 | 14.33% (4.49%-37.31%) | 67.30% |
| **2000–2004** | 7 | 9.33% (3.89%-20.73%) | 81.30% | 6 | 5.61% (2.01%-14.70%) | 74.20% | | 28% (21%-38%) | 0% | - | - | - |
| **2005–2009** | 5 | 13.63% (8.94%-20.24%) | 63.60% | 6 | 12.87% (6.29%-24.52%) | 98% | | 15% (6%-32%) | 96% | 4 | 17.47% (8.93%-31.36%) | 81.90% |
| **2010–2014** | 13 | 10.68% (6.53%-16.98%) | 82.90% | 7 | 10.80% (4.18%-25.11%) | 92.20% | | 28% (16%-45%) | 66% | 4 | 24.03% (7.50%-55.23%) | 93.50% |
| **2015–2019** | 3 | 17.11% (5.52%-42.15%) | 83.50% | 4 | 5.05% (2.35%-10.51%) | 71.70% | 1 | 22% (15%-30%) | NA | 1 | 22.73% (14.19%-34.34%) | NA |
| **by age** | | | | | | | | | | | | |
| **0–5 years** | 15 | 7.18% (4.31%-11.70%) | 93.90% | 14 | 4.88% (3.49%-6.79%) | 60.90% | 6 | 28% (21%-37%) | 71% | 2 | 10.21% (1.51%-45.79%) | 95.70% |
| **6–64 years** | 2 | 15.95% (3.69%-48.44%) | 87.80% | 2 | 16.84% (6.59%-36.78%) | 75.90% | 1 | 14% (7%-37%) | NA | 2 | 24.47% (7.87%-55.12%) | 76% |
| **≥ 65 years** | 2 | 44.99% (28.50%-62.66%) | 76.50% | 2 | 29.67% (6.80%-70.92%) | 91.50% | - | - | - | 1 | 66.67% (42.88%-84.20%) | NA |
| **by country** | | | | | | | | | | | | |
| **Argentina** | 5 | 4.56% (2.33%-8.73%) | 71.70% | 8 | 5.98% (2.87%-12.04%) | 87.80% | 4 | 15% (8%-24%) | 62% | 4 | 11.72% (6.89%-19.24%) | 76.70% |
| **Brazil** | 5 | 17.86% (10.45%-28.85%) | 88.00% | 2 | 4.37% (1.91%-9.66%) | 85.80% | 13 | 22% (16%-30%) | 99% | 2 | 19.44% (6.57%-45.30%) | 82.90% |
| **Chile** | 3 | 5.97% (3.33%-10.47%) | 11.10% | 4 | 29.31% (21.88%-38.04%) | 31.30% | 2 | 14% (11%-18%) | 0% | 1 | 8.88% (6.45%-12.12%) | NA |
| **Colombia** | 5 | 26.01% (14.65%-41.86%) | 92.90% | 6 | 13.30% (5.59%-28.44%) | 95.30% | 5 | 29% (21%-38%) | 25% | 3 | 33.08% (9.94%-68.90%) | 89.30% |
| **Costa Rica** | 1 | 14.39% (9.37%-21.47%) | NA | 1 | 22.22% (11.52%-38.53%) | NA | 1 | 16% (8%-28%) | NA | 1 | 6.67% (1.67%-23.07%) | NA |
| **Cuba** | 3 | 6.59% (2.33%-17.26%) | 61.20% | - | - | - | 2 | 29% (20%-40%) | 94% | - | - | - |
| **Ecuador** | 1 | 5.17% (4.62%-5.78%) | NA | - | - | - | - | - | - | - | - | - |
| **French-Guiana** | - | - | - | - | - | - | 1 | 33% (16%-55%) | NA | - | - | - |

*(Continued)*

**Table 2.** (Continued)

| Case fatality rate (CFR) | IPD | | | Pneumonia | | | Meningitis | | | Bacteremia | | |
|---|---|---|---|---|---|---|---|---|---|---|---|---|
| | Studies number | Proportion (CI 95%) | I2 | Studies number | Proportion (CI 95%) | I2 | Studies number | Proportion (CI 95%) | I2 | Studies number | Proportion (CI 95%) | I2 |
| Guatemala | 2 | 21.40% (18.52%-24.59%) | 0% | 1 | 2.30% (0.87%-5.96%) | NA | 2 | 29% (16%-47%) | 89% | 2 | 23.34% (17.25%-30.77%) | 0% |
| Jamaica | 1 | 9.23% (4.21%-19.06%) | NA | - | - | - | 1 | 12% (3%-31%) | NA | 1 | 8.16% (3.10%-19.82%) | NA |
| Mexico | 5 | 25.40% (21.55%-29.67%) | 46.20% | 1 | 16.00% (6.14%-35.69%) | NA | 1 | 38% (23%-54%) | NA | 1 | 31.08% (21.61%-42.46%) | NA |
| Panama | 1 | 30.43% (15.25%-51.54%) | NA | - | - | - | - | - | - | - | - | - |
| Paraguay | 1 | 6.41% (2.69%-14.49%) | NA | - | - | - | 1 | 41% (27%-57%) | NA | - | - | - |
| Peru | 4 | 18.18% (14.11%-23.12%) | 1.10% | 2 | 11.40% (6.86%-18.34%) | 0% | 2 | 77% (18%-36%) | 0% | 1 | 25.00% (6.30%-62.29%) | NA |
| Puerto Rico | 1 | 17.71% (12.93%-23.76%) | NA | - | - | - | - | - | - | - | - | - |
| Uruguay | 3 | 11.46% (6.43%-19.59%) | 67.10% | 8 | 5.74% (3.32%-9.73%) | 85.10% | 4 | 28% (19%-38%) | 72% | - | - | - |

impact in Chile and Colombia, when we analyze specific serotypes included in PCVs. The serotype distribution, including PCV20 in Argentina, Brazil, Chile, and Colombia, is shown in Fig 4 **Percentage of *S. pneumoniae* serotypes in passive surveillance included in PCV10 and PCV13 in Argentina, Brazil, Chile, and Colombia between 2006 and 2018 and the percentage of PCV20 serotypes between 2013 and 2018**.

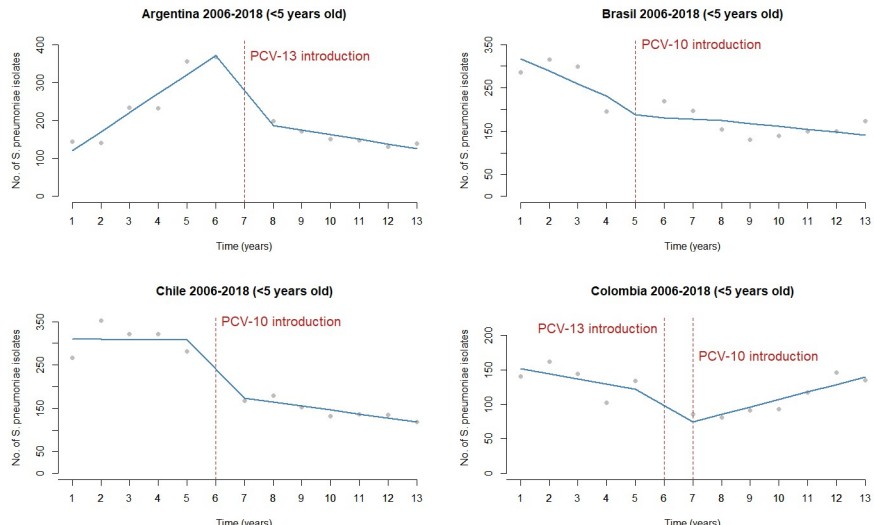

**Fig 3. Interrupted time series of *S. pneumoniae* isolates before and after PCV13 and PCV10 introduction in Argentina, Brazil, Chile, and Colombia from 2006 to 2018.** The red dotted line indicates the year of introduction of the vaccines.

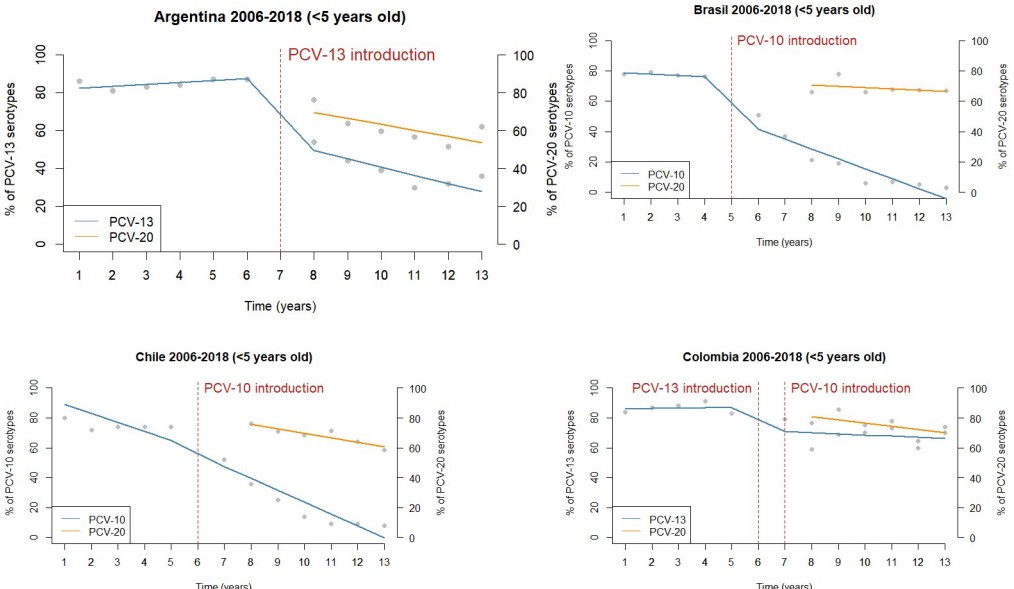

**Fig 4. Interrupted time series of the percentage of *S. pneumoniae* serotypes in passive surveillance included in PCV10 and PCV13 in Argentina, Brazil, Chile, and Colombia between 2006 and 2018 and the percentage of PCV20 serotypes (orange dotted line) between 2013 and 2018.** The red dotted line indicates the year of introduction of the vaccines.

S4 Fig 1 in S3 File shows the percentage of serotypes by vaccine type, non-vaccine, and non-typable isolates per year and country. Selected data from 6 countries with over 2,500 IPD isolates of all ages is reported by the number of isolates per 100,000 inhabitants from Argentina, Brazil, Chile, Colombia, Mexico, and Uruguay. S4 Fig 2 in S3 File shows the number of isolates per 100,000 population, 2006–2018 in seven LAC countries by age groups. In these countries, a significant decreasing trend in vaccine serotypes and an increase in non-vaccine types were observed after the introduction of PCVs. Due to this emergence of non-vaccine types, serotype replacement occurs, and the changes observed in each country showed a correlation with the pneumococcal conjugate vaccines used. S4 Fig 3 in S3 File shows vaccine serotypes decreased after introducing PCV10 or PCV13 vaccines in seven countries. To compare the situation in the different countries, the year of vaccine introduction was considered time zero. To depict the changes that occurred after the incorporation of PCVs, we analyzed data from the latest SIREVA global report available from the countries that in 2018 reported more than 100 IPD isolates in children under six. The prevalent non-vaccine types were 12F and 24F in Argentina, where PCV13 was introduced in 2012; 19A, 3, and 24F in Brazil and Chile, where PCV10 was used from 2010 and 2011, respectively; and 19A and 6C in Colombia, where PCV13 was introduced in 2011 and replaced by PCV10 one year later.

**Risk of bias assessment.** Among 83 cross-sectional studies, eight cohort studies, and three ecological studies. 55 (60%) were evaluated as carrying a moderate risk of bias, 26 (29%) a low risk of bias, and ten studies (11%) a high risk of bias. The domains that presented the most significant inconvenience in their evaluation were those related to sample size calculation and the measurements of the results according to different exposures. Likewise, other drawbacks in the assessment were associated with the validation domains of the results regarding the blinding of the observers and the adjustment for possible confounding variables. Of note, other domains could not be assessed due to the nature of the studies, such as repeated

measurements and follow-up in cross-sectional studies. S3 Table 4 in S2 File. For the 61 case series, 26 studies (65%) had a low risk of bias, 20 (33%) were moderate, and only one study (2%) was evaluated with a high risk of bias. The major inconveniences were defining whether the cases included were consecutive and identifying issues related to statistical information management. S3 Table 5 in S2 File.

## Discussion

We included 155 epidemiological studies on *S. pneumoniae* published in 2000–2022 in 22 LAC countries. Studies provided data on invasive disease, meningitis, pneumonia, and serotypes across pediatric, adult, and mixed populations in healthcare facilities. Reported epidemiological data spanned 1977–2018, with increasing studies in the past two decades. Key findings included serotype distribution, antibiogram profiles, and regional disease burden.

We found a summary proportion of pneumococcal serotypes included in PCV10 and PCV13 similar to other reported studies regarding global IPD and their different clinical manifestations [14, 185]. Data collected since 2006 by the SIREVA surveillance network in LAC showed a decrease in vaccine serotypes and an increase in non-vaccine serotypes in IPD after the introduction of PCV10/13. Countries with > 2,500 isolates from children under five between 2006 and 2018, like Argentina, Brazil, Chile, Colombia, Mexico, and Uruguay, showed significant serotype replacement. Post-PCV13, serotypes 12F, 24F, 19A, 3, 6C emerged. New higher-valent vaccines like PCV20 could improve coverage, reaching a theoretical coverage between 62 and 70% for strains isolated in these countries during 2018. Considering these findings, ongoing surveillance is needed to monitor serotype dynamics and guide optimal PCV formulation. National passive surveillance systems such as SIREVA are valuable tools to monitor serotype distribution and antimicrobial resistance but do not provide disease burden data, so they are unsuitable for evaluating pneumococcal vaccines' impact and could underestimate IPD incidence [186].

Other studies found that introducing PCVs in the LAC region significantly reduced IPD caused by vaccine serotypes in children under five. Specifically, the study found that the proportion of IPD caused by vaccine serotypes decreased from 57.3% before PCV introduction to 6.5% after PCV introduction [187]. This reduction was accompanied by an increase in the proportion of non-vaccine serotypes causing IPD. These findings contribute to our understanding of the effectiveness of PCVs in preventing invasive pneumococcal disease in children and highlight the importance of ongoing surveillance to monitor changes in serotype distribution over time [8].

Another study found that introducing the PCV10 in the Brazilian National Immunization Program significantly reduced IPD among hospitalized children under five. The vaccine effectively prevented IPD caused by vaccine serotypes and reduced the need for intensive care unit admission and the mortality rate associated with IPD [61]. In some countries where PCV10 was introduced, like Chile, Colombia, Belgium, and Brazil, an increase in serotypes not targeted by the vaccine, particularly 19A serotype related to penicillin and multidrug resistance, has been observed [64].

In Argentina, Zintgraff et al. analyzed the epidemiology of invasive pneumococcal disease in children under five from 2006–2019. After the introduction of PCV13 in 2012, vaccine serotypes like 14, 5, and 1 declined while non-vaccine serotypes, including 12F and 24F, increased. Serotype 19A persisted at ~5% prevalence despite vaccination. Antimicrobial resistance increased post-PCV13, now driven by non-vaccine serotypes [188].

While introducing pneumococcal conjugate vaccines in LAC has led to declines in vaccine-type IPD, serotype replacement with increases in non-vaccine serotypes has occurred. Passive

laboratory surveillance has limitations in determining true disease incidence and vaccine impacts. Ongoing surveillance and higher-valent vaccines are crucial to face the remaining pneumococcal disease burden. However, gaps exist in the knowledge of serotype-specific disease incidence, indirect vaccine effects, and optimal strategies for broader protection. A worldwide systematic review was conducted to identify studies and surveillance reports (published between 2000 and December 2015) of pneumococcal serotypes causing childhood IPD after PCV introduction [11]. The main findings of this review were that after the introduction of PCVs, a considerable proportion of childhood IPD was caused by non-PCV13 serotypes and, among the vaccine serotypes, 19A was the most frequently identified in different regions of the world. A recent review of surveillance systems or hospital networks in children from high-income countries found that IPD of serotype PCV13 caused 37.4% of total IPD cases. Including next-generation PCVs in existing pediatric programs may reduce the incidence of IPDs [189]. Serotype distribution studies were also published for Sweden [190] Spain [191], Portugal [192], Argentina [39], Kuwait [193], South Africa [194], India [195], and Europe [196].

Developing new conjugate vaccines that include these emerging serotypes will likely improve the situation. The recently approved PCV20 adds serotypes 8, 10A, 11A, 12F, 15B/15C, 22F and 33F to the PCV13 serotypes. SIREVA data from 2018 of IPD in children less than five years in four selected countries showed that the proportion of serotypes coverage was: PCV10 16%, PCV13 36%, and PCV20 62% in Argentina; PCV10 3%, PCV13 55% and PCV20 67% in Brazil; PCV10 8%, PCV13 39% and PCV20 59% in Chile and PCV10 7%, PCV13 69% and PCV20 70% in Colombia.

IPD in the region in the last years was prevalent in children under five and adults over 65. Also, an increase in non-vaccine serotypes was observed in the late period after the introduction of PCV10 and PCV13.

Our study has many strengths, uncovering all published and unpublished data in 22 LAC countries. These studies covered diverse aspects of *S. pneumoniae*, such as invasive disease, meningitis, and pneumonia, spanning different age groups and healthcare facilities. The dataset extended from 1977 to 2018, reflecting a growing body of research over the past two decades and analyzing passive surveillance. Our key findings revolved around serotype distribution, antibiogram profiles, and regional disease burden. As a limitation, sample selection, size, and outcome measurement were the main issues when assessing bias risk. Case series were mostly low-risk (63.5%) or moderate (35%), with selection bias and statistical methods concerns.

In summary, pneumococcal conjugate vaccines significantly reduced IPD and shifted serotype distribution in Latin America and the Caribbean. PCV10/PCV13 covered 57–84% of serotypes in children under 5, with marked decline in PCV serotypes post-vaccination. Our findings contribute, jointly with continuous surveillance, to addressing regional changes in pneumococcal serotypes in IPD and to analyzing epidemiological and clinical data before introducing a new pneumococcal vaccine to LAC countries.

## Supporting information

**S1 Checklist. PRISMA checklist.**
(DOCX)

**S1 File. Search strategy.**
(DOCX)

**S2 File.**
(DOCX)

**S3 File.**
(DOCX)

## Acknowledgments

The authors thank Mr. Daniel Comande, the Institute for Clinical Effectiveness and Health Policy librarian, for contributing to the bibliographic searches.

## Author Contributions

**Conceptualization:** Ariel Bardach, Silvina Ruvinsky, Agustín Ciapponi.

**Data curation:** M. Carolina Palermo, Tomás Alconada, M. Macarena Sandoval, Martín E. Brizuela.

**Formal analysis:** Silvina Ruvinsky, Tomás Alconada, Joaquín Cantos, Paula Gagetti, Agustín Ciapponi.

**Funding acquisition:** Ariel Bardach, Agustín Ciapponi.

**Investigation:** Ariel Bardach, Silvina Ruvinsky, M. Carolina Palermo, Tomás Alconada, M. Macarena Sandoval, Martín E. Brizuela, Eugenia Ramirez Wierzbicki, Paula Gagetti, Agustín Ciapponi.

**Methodology:** Ariel Bardach, Silvina Ruvinsky, Paula Gagetti, Agustín Ciapponi.

**Project administration:** Ariel Bardach, Agustín Ciapponi.

**Resources:** Ariel Bardach, Silvina Ruvinsky, Agustín Ciapponi.

**Supervision:** Ariel Bardach, Silvina Ruvinsky, Paula Gagetti, Agustín Ciapponi.

**Validation:** Paula Gagetti.

**Writing – original draft:** Ariel Bardach, Silvina Ruvinsky, M. Carolina Palermo, Tomás Alconada, M. Macarena Sandoval, Martín E. Brizuela, Eugenia Ramirez Wierzbicki, Joaquín Cantos, Paula Gagetti, Agustín Ciapponi.

**Writing – review & editing:** Ariel Bardach, Silvina Ruvinsky, M. Carolina Palermo, Tomás Alconada, M. Macarena Sandoval, Martín E. Brizuela, Eugenia Ramirez Wierzbicki, Joaquín Cantos, Paula Gagetti, Agustín Ciapponi.

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
