## [Decision Letter · Decision Letter 0]

19 Jan 2024

PONE-D-23-31108Invasive pneumococcal disease in Latin America and the Caribbean: serotype distribution, disease burden, and impact of vaccination. A systematic review and meta-analysisPLOS ONE

Dear Dr. Bardach,

Thank you for submitting your manuscript to PLOS ONE. After careful consideration, we feel that it has merit but does not fully meet PLOS ONE’s publication criteria as it currently stands. Therefore, we invite you to submit a revised version of the manuscript that addresses the points raised during the review process.

We look forward to receiving your revised manuscript.

Kind regards,

Anirudh K. Singh, Ph.D

Academic Editor

PLOS ONE

“Pfizer Independent Grant 76436251”

3. Please include a separate caption for each figure in your manuscript

Reviewers' comments:

Reviewer's Responses to Questions

**Comments to the Author**

1. Is the manuscript technically sound, and do the data support the conclusions?

Reviewer #1: Yes

Reviewer #2: No

Reviewer #3: Yes

2. Has the statistical analysis been performed appropriately and rigorously? 

Reviewer #1: Yes

Reviewer #2: Yes

Reviewer #3: I Don't Know

3. Have the authors made all data underlying the findings in their manuscript fully available?

Reviewer #1: Yes

Reviewer #2: Yes

Reviewer #3: Yes

4. Is the manuscript presented in an intelligible fashion and written in standard English?

Reviewer #1: Yes

Reviewer #2: Yes

Reviewer #3: Yes

5. Review Comments to the Author

Reviewer #1: The study's strengths lie in its diverse coverage of S. pneumoniae aspects across multiple countries and healthcare facilities. However, the study transparently addresses limitations related to sample selection, size, and outcome measurement, acknowledging potential bias risks in data assessment. In summary, the study provides valuable insights into pneumococcal serotype distribution in Latin American IPD. While acknowledging the positive impact of PCVs in reducing vaccine serotypes, it underscores the ongoing challenge of serotype replacement. The study's call for continuous surveillance to monitor serotype distribution changes and address the persisting disease burden is well-founded, making it a pertinent contribution to the field. 

Reviewer #2: The authors present a comprehensive secondary analysis taking in the analysed data out come of various reports published in period of time. The analysis might be helpful provided the data taken to be rigorous and analysed on various statistical parameters.

-Please discuss the outcome of the study in detail incorporating molecular mechanisms based on those studies have had reported.

-Please move table 1 to supplementary section.

-Think about more suitable representation of table 2 and 3, you can think of segmentation of these respective figures.

-Discuss a little about he analysis algorithms that you are following.

-A structured discussion incorporating above suggestion will present your views more appropriately.

Reviewer #3: The review is important for the field and the countries included in this review are not well studied. I would recommend the authors to refine the manuscript to target larger audience.

These references should be referred to express the key findings of a systematic review and meta-analysis.

DOI: 10.1016/j.tmaid.2018.05.016

DOI: 10.1016/S2214-109X(16)30306-0

I have few other comments which are as follows:

Abstract:

• Line 35: “death hospitalization” can be replaced with death after hospitalization or similar.

• Line 36: correct serotypes IPD to serotyped IPD.

• Line 37: the first line of the result should be shifted to the method section. Or should be reworded to say that 155 studies were screened etc.

• Line 38: if invasive diseases mean IPD, then it IPD should be used consistently.

• Include some result about the overall incidences in different age groups and the few key findings.

• Line 42: specify countries from the LAC region.

Introduction:

• Line 52: use some other word instead of sterile materials.

• Line 55-56: The sentence should be reworded for better clarity.

• Line 65: Include the full form of SIREVA.

• Line 70: The sentence needs to be reworded to say that that serotypes of the vaccines have been reduced in the population due to vaccine 7/10/13 implantation.

The methodology looks elaborate and well written which needs to be checked by the experts in the field of meta-analysis.

Results:

• Table 1: The outcomes column should be populated more to include serotypes, incidences etc.

Discussion:

• Line 316: Used the abbreviation IPD consistently.

• Line 334-336: Highlight some key finding from the worldwide study.

• Line 347: Write the region name or use LAC consistently.

• The conclusion should be in line with the title of the review.

6. PLOS authors have the option to publish the peer review history of their article (what does this mean?). If published, this will include your full peer review and any attached files.

Reviewer #1: No

Reviewer #2: No

Reviewer #3: No

---

## [Author Response · Author response to Decision Letter 0]

16 Feb 2024

Response to Reviewers:

2. Thank you for stating the following financial disclosure: “Pfizer Independent Grant 76436251” Please state what role the funders took in the study. If the funders had no role, please state: "The funders had no role in study design, data collection and analysis, decision to publish, or preparation of the manuscript." If this statement is not correct you must amend it as needed. Please include this amended Role of Funder statement in your cover letter; we will change the online submission form on your behalf.

Thanks, we already rephrased it in the manuscript and it was modified and highlighted in the cover letter.

3. Please include a separate caption for each figure in your manuscript

Done

All file captions were added at the end and updated in the manuscript.

Reviewer #1: The study's strengths lie in its diverse coverage of S. pneumoniae aspects across multiple countries and healthcare facilities. However, the study transparently addresses limitations related to sample selection, size, and outcome measurement, acknowledging potential bias risks in data assessment. In summary, the study provides valuable insights into pneumococcal serotype distribution in Latin American IPD. While acknowledging the positive impact of PCVs in reducing vaccine serotypes, it underscores the ongoing challenge of serotype replacement. The study's call for continuous surveillance to monitor serotype distribution changes and address the persisting disease burden is well-founded, making it a pertinent contribution to the field. 

Many thanks for your review.

Reviewer #2: The authors present a comprehensive secondary analysis taking in the analyzed data outcome of various reports published in a period of time. The analysis might be helpful provided the data taken to be rigorous and analyzed on various statistical parameters.

-Please discuss the outcome of the study in detail incorporating molecular mechanisms based on those studies that have been reported.

Molecular methods were not described in detail because this is not one of the objectives of the present work, which describes mostly the epidemiology of serotypes in the region. In addition, only a handful of the studies included this detail. 

-Please move table 1 to supplementary section.

Thank you for your comment. However, the group of authors considers that Table 1 summarizes well the main characteristics of the evidence identified for the region and is the initial type of table for any epidemiological systematic review. As such, we feel it should be included in the main body of the manuscript. Sacrificing this content detracts from the value of the work. 

-Think about a more suitable representation of table 2 and 3, you can think of segmentation of these respective figures.

Thank you for your comment. Table 2 shows the detailed results of proportion meta-analyses of the distribution of vaccine serotypes by condition in LAC, by five-year period, age groups, and countries. In the opinion of the authors' group, it was a very thoughtful and difficult table to produce, and we think it is very informative. We feel it does not require further revamping, as relevant categories have already segmented information. Table 3 shows the Case fatality rate according to type of IPD and sub-analysis by 5-year period, age, and country. This important health outcome accounts for the virulence of circulating serotypes in invasive pneumococcal disease and allows compatibility with other regions. We believe it adds real value to the study.

-Discuss a little about the analysis algorithms that you are following.

Thank you. We are following the PRISMA statement algorithm for systematic reviews (cited in the text), including the study flowchart (Fig 1), generated by the Covidence Software, as explained in the manuscript. The search algorithm is also explained in the Methods section.

-A structured discussion incorporating the above suggestion will present your views more appropriately.

Thank you. Several changes have been introduced to improve the structure and readability.

Reviewer #3: The review is important for the field and the countries included in this review are not well studied. I would recommend the authors to refine the manuscript to target a larger audience.

These references should be referred to express the key findings of a systematic review and meta-analysis.

DOI: 10.1016/j.tmaid.2018.05.016

DOI: 10.1016/S2214-109X(16)30306-0

Thanks for your suggestion. We incorporated both references in the introduction section and were highlighted in the references.

I have few other comments which are as follows:

Abstract:

• Line 35: “death hospitalization” can be replaced with death after hospitalization or similar. Done

• Line 36: correct serotypes IPD to serotyped IPD. 

Done

• Line 37: the first line of the result should be shifted to the method section. Or should be reworded to say that 155 studies were screened etc. 

Done

• Line 38: if invasive diseases mean IPD, then it IPD should be used consistently. 

Modified. The same acronym is used throughout the manuscript.

• Include some results about the overall incidences in different age groups and the few key findings. 

This is described in the text, in lines 235-243 and S3 Table and Figure 2.

• Line 42: specify countries from the LAC region. 

All countries of Latin America and the Caribbean were included, as defined by the United Nations 

Introduction:

• Line 52: use some other word instead of sterile materials. 

Modified

• Line 55-56: The sentence should be reworded for better clarity. 

Done

• Line 65: Include the full form of SIREVA. 

The Spanish meaning was clarified.

• Line 70: The sentence needs to be reworded to say that serotypes of the vaccines have been reduced in the population due to vaccine 7/10/13 implantation. 

Done

The methodology looks elaborate and well written which needs to be checked by the experts in the field of meta-analysis.

Thanks

Results:

• Table 1: The outcomes column should be populated more to include serotypes, incidences etc. 

Thank you. 

Table 1 is a summary table of the overall type of available evidence in the region. Vaccine serotypes in LAC evidence are meta-analyzed and further summarized in Table 2. Incidence meta-analyses conducted are described in the text (lines 235-247) of Results. The author's group feels that including such information in Table 1 would make it even larger and redundant.

Discussion:

• Line 316: Used the abbreviation IPD consistently. 

Done

• Line 334-336: Highlight some key findings from the worldwide study. 

Done. To clarify a paragraph was added as follows: The main findings of this review were that after the introduction of PCVs, a considerable proportion of childhood IPD was caused by non-PCV13 serotypes and, among the vaccine serotypes, 19A was the most frequently identified in different regions of the world.

• Line 347: Write the region name or use LAC consistently. 

LAC was consistently placed in the manuscript. 

• The conclusion should be in line with the title of the review. 

Thanks, we re-wrote the conclusions to be more in line with the title.

---

## [Editor Report · Decision Letter 1]

25 Mar 2024

PONE-D-23-31108R1Invasive pneumococcal disease in Latin America and the Caribbean: serotype distribution, disease burden, and impact of vaccination. A systematic review and meta-analysisPLOS ONE

Dear Dr. Bardach,

Thank you for submitting your manuscript to PLOS ONE. After careful consideration, we feel that it has merit but does not fully meet PLOS ONE’s publication criteria as it currently stands. Therefore, we invite you to submit a revised version of the manuscript that addresses the points raised during the review process.

Please follow reviewer# 2 suggestion to move Table 1 to supplementary information. Please read PlosOne policy on reference/citations. I am not sure you can cite a study/abstract/poster if it is not availabe in the public domain.

We look forward to receiving your revised manuscript.

Kind regards,

Anirudh K. Singh, Ph.D

Academic Editor

PLOS ONE

Journal Requirements:

Additional Editor Comments:

Please follow reviewer# 2 suggestion to move Table 1 to supplementary information. Please read PlosOne policy on reference/citations. I am not sure you can cite a study/abstract/poster if it is not availabe in the public domain.

---

## [Author Response · Author response to Decision Letter 1]

28 Mar 2024

Response to Reviewers R2:

Please follow reviewer# 2 suggestion to move Table 1 to supplementary information. 

Thank you. Done

Please read PlosOne policy on reference/citations. I am not sure you can cite a study/abstract/poster if it is not availabe in the public domain.

Thank you, we added more information and links to these 3 references:

Benitez JD, Martinez ME, Specht MH von, Gerlach E, Gonzalez CA, Grenon SL. Epidemiology and risk factors for invasive pneumococcal disease in pediatrics. Descriptive, postvaccinal study. Epidemiología y factores de riesgo de enfermedad invasiva neumocócica en pediatría Estudio descriptivo, postvacunal. Revista de ciencia y tecnologia 2017; 4–10. Available: http://ri.conicet.gov.ar/handle/11336/177141 Accessed May 5th 2023

Agudelo CI, Sanabria OM, Ovalle MV, Castaneda E. Laboratory surveillance of Streptococcus pneumoniae serotypes from infected children aged less than 5 years: 1994-2000. Biomédica 2001;21(2): 193–199.

Agudelo I, Diaz LP, Sanabria MO, Ovalle VM, Castaneda E, Rosa Gallego C, et al. Laboratory surveillance of Streptococcus pneumoniae isolated by invasive processes in the population older than 5 years, 1998-2001. Vigilancia por el laboratorio de Streptococcus pneumoniae aislado de procesos invasores en población mayor de 5 años, 1998-2001. 2002;7: 177–183. https://www.cabidigitallibrary.org/doi/full/10.5555/20023161394

Accessed May 5th 2023

---

## [Editor Report · Decision Letter 2]

9 Apr 2024

Invasive pneumococcal disease in Latin America and the Caribbean: serotype distribution, disease burden, and impact of vaccination. A systematic review and meta-analysis

PONE-D-23-31108R2

Dear Dr. Bardach,

We’re pleased to inform you that your manuscript has been judged scientifically suitable for publication and will be formally accepted for publication once it meets all outstanding technical requirements.

Kind regards,

Anirudh K. Singh, Ph.D

Academic Editor

PLOS ONE
---

## [Editor Report · Acceptance letter]

28 May 2024

PONE-D-23-31108R2 

PLOS ONE

Dear Dr. Bardach, 

I'm pleased to inform you that your manuscript has been deemed suitable for publication in PLOS ONE. Congratulations! Your manuscript is now being handed over to our production team.

Kind regards, 

on behalf of

Dr. Anirudh K. Singh 

Academic Editor

PLOS ONE